# Potential Exposure to Respiratory and Enteric Bacterial Pathogens among Wastewater Treatment Plant Workers, South Africa

**DOI:** 10.3390/ijerph20054338

**Published:** 2023-02-28

**Authors:** Evida Poopedi, Tanusha Singh, Annancietar Gomba

**Affiliations:** 1National Institute for Occupational Health, National Health Laboratory Service, Johannesburg 2000, South Africa; 2Department of Clinical Microbiology and Infectious Diseases, University of the Witwatersrand, Johannesburg 2050, South Africa; 3Department of Environmental Health, University of Johannesburg, Doornfontein 2028, South Africa

**Keywords:** domestic wastewater, sanitation workers, biological hazards, occupational exposure, waterborne diseases

## Abstract

Wastewater handling has been associated with an increased risk of developing adverse health effects, including respiratory and gastrointestinal illnesses. However, there is a paucity of information in the literature, and occupational health risks are not well quantified. Grab influent samples were analysed using Illumina Miseq 16S amplicon sequencing to assess potential worker exposure to bacterial pathogens occurring in five municipal wastewater treatment plants (WWTPs). The most predominant phyla were *Bacteroidota*, *Campilobacterota*, *Proteobacteria*, *Firmicutes*, and *Desulfobacterota*, accounting for 85.4% of the total bacterial community. Taxonomic analysis showed a relatively low diversity of bacterial composition of the predominant genera across all WWTPs, indicating a high degree of bacterial community stability in the influent source. Pathogenic bacterial genera of human health concern included *Mycobacterium*, *Coxiella*, *Escherichia/Shigella*, *Arcobacter*, *Acinetobacter*, *Streptococcus*, *Treponema*, and *Aeromonas*. Furthermore, WHO-listed inherently resistant opportunistic bacterial genera were identified. These results suggest that WWTP workers may be occupationally exposed to several bacterial genera classified as hazardous biological agents for humans. Therefore, there is a need for comprehensive risk assessments to ascertain the actual risks and health outcomes among WWTP workers and inform effective intervention strategies to reduce worker exposure.

## 1. Introduction

Municipal wastewater naturally contains hazardous biological agents, including bacteria, fungi, and viruses [1]. A significant amount of municipal wastewater is generated by households and contains human excreta such as faeces and urine, including that from disease-carrying individuals. Waterborne infections and diseases caused by ingestion, inhalation, or dermal contact with hazardous biological agents remain a global public health concern [1]. Daily work activities performed by WWTP workers such as manual screening and desludging, plant maintenance, and emergency repairs increase the risk of exposure to preventable waterborne diseases [2]. Although the World Health Organization (WHO) recognises the occupational health risks of WWTP workers [2], specific guidelines to protect wastewater workers from workplace hazards remain elusive, particularly in low to middle-income countries (LMICs). Consequentially, workers at wastewater treatment plants (WWTPs) are at perpetual risk of exposure to waterborne pathogens through different exposure routes [3]. Higher prevalence of gastrointestinal [4,5,6] and respiratory [6,7,8,9] symptoms have been reported in WWTP workers compared to controls. For instance, in a study to investigate the impact of inhalable particles and gas exposure on the respiratory system of WWTP workers from urban and rural sewage plants and the sewer net system in Norway, Heldal and co-workers (2019) [9] observed a lower lung function and a higher prevalence of airway symptoms (33 and 11%, respectively) among WWTP workers compared to the control group [8]. Furthermore, Van Hooste et al. (2010) [4] reported a higher prevalence (37%) of gastrointestinal symptoms such as stomach ache, abdominal pain, indigestion, heartburn, and burping compared to the control group (14.7%) among WWTP workers in Belgium [4]. However, the precise cause of the reported symptoms and illnesses is not well studied.

Among the most commonly detected microbial contaminants in municipal wastewater with potential to cause health issues in exposed populations, bacteria and their components are a major concern, particularly for immunocompromised individuals [10]. While a lot of studies on wastewater bacteriology employ traditional culture tests and/or targeted molecular assays to provide absence/presence and quantified measurements of a few selected bacteria, the advent of advanced high throughput DNA sequencing technologies dramatically increases the scope to comprehensively identify even the fastidious microorganisms. Therefore, advanced sequencing technologies provide a more accurate representation of the microbial population in a given sample with high taxonomic resolution [11].

In recent years, researchers have employed high throughput molecular approaches such as targeted amplicon sequencing and shotgun metagenomics to profile bacterial communities in wastewater and sludge. These studies have revealed the presence of several pathogenic bacteria at community level, including *Streptococcus pneumoniae*, *Vibrio cholerae*, *Salmonella* spp., *Mycobacterium tuberculosis*, *Legionella pneumophila*, *Pseudomonas aeruginosa*, *Arcobacter* spp., and *Acinetobacter* spp., among others [10,12,13,14,15]. Considering that WWTP workers are potentially exposed to a plethora of harmful bacterial genera at any given time, knowledge of occupational exposure risks at microbial community level is critical. Moreover, microorganisms have evolved together in communities, and certain health effects may be due to synergistic or antagonistic behaviors of specific pathogens [16]. Additionally, understanding microbial composition at community level enables microbial source tracking and comprehensive risk assessment to minimise exposure to harmful biological agents.

Microbial communities in municipal wastewater and the extent of worker exposure may differ depending on geographic location, season, time of the day, population served, facility treatment capacity, the technology used, and performed activities [17,18,19]. Yet there is a general lack of research in LMICs to characterise bacterial communities in untreated wastewater using high throughput sequencing technologies [15,20,21], with little to no data documented in most regions [22,23]. In South Africa, for example, previous studies in wastewater environments have mainly focused on pathogen removal and effluent quality, essential to protect public health and the environment. Consequently, the likelihood of occupational exposure to untreated or partially treated wastewater at WWTPs and the associated health risks in LMICs remain unclear. Hence, in this study we used targeted 16 rRNA gene amplicon sequencing to describe bacterial diversity and identify bacterial pathogens that could pose an occupational health risk from repeated exposure to untreated municipal wastewater.

## 2. Materials and Methods

### 2.1. Study Design

The study was an experimental design conducted at the National Institute for Occupational Health, Immunology and Microbiology Laboratory, South Africa. Grab influent samples were obtained from five participating WWTPs on 20 October (WWTP2, 3 and 4) and 21 October (WWTP1 and 5), 2020.

### 2.2. Site Description

Sampling was conducted at five municipal WWTPs in the City of Tshwane Metropolitan Municipality, South Africa. The characteristics of the sampling sites are summarised in Table 1. The selected WWTPs all receive wastewater from healthcare facilities; however, the quantities are not measured. Non-sewered informal settlements were observed nearby WWTP1, 2, and 4 but no intensive animal (cattle or battery chicken) production or roaming domestic animals were evident near any of the sampling sites.

### 2.3. Sample Collection

Grab influent samples were collected in 1 litre sterile polypropylene bottles (Sigma-Aldrich, Saint Louis, MO, USA). Samples were transported on ice and stored at 2–8 °C for not more than 24 h until analysis. Sample temperature and pH were measured during sampling using a digital water-resistant thermometer 125 mm (−50 to +200 °C) (Lasec, Cape Town, South Africa). Metadata such as environmental temperature, relative humidity, and CO_2_ were measured using IAQ-CALC Indoor Air Quality Meter, Model 7545 (TSI Instruments Ltd., High Wycombe, UK)

### 2.4. Microbial Cell Concentration

Microbial cells were concentrated following the protocol by Kumar et al. (2020) [24]. Briefly, each 1 L grab sample was manually shaken by hand to allow adequate mixing and an aliquot of 200 mL in four batches of 50 mL was centrifuged at 5000 rpm for 45 min at 4 °C. The supernatant was carefully removed and each pellet was washed in 1 mL of sterile distilled water to remove deposited salts and other impurities. The four pellets were pooled and stored at −20 °C until further analysis.

### 2.5. Total Genomic DNA Extraction and 16 S Sequencing

Total genomic DNA (gDNA) was extracted from the pelleted biomass of influent samples using DNeasy Powersoil Kit (Qiagen, Hilden, Germany) according to the manufacturer’s instructions. The DNA was quantified using a Nanodrop 2000c spectrophotometer (Thermo Fisher Scientific, Waltham, MA, USA). The gDNA was amplified with primers targeting bacterial V3 and V4 regions, namely, 16S Amplicon PCR forward primer (5′-TCG TCG GCA GCG TCA GAT GTG TAT AAG AGA CAG CCT ACG GGN GGC WGC AG) and 16S Amplicon PCR reverse primer (5′-GTC TCG TGG GCT CGG AGA TGT GTA TAA GAG ACA GGA CTA CHV GGG TAT CTA ATC C) [25]. The PCR reaction contained 12.5 µL of 2x KAPA HiFi HotStart ready mix, 5 ng/µL gDNA template, and 5 µL of 1 µM each primer. Cycling conditions included initial denaturation at 95 °C for 3 min, 25 cycles of denaturation at 95 °C for 30 s, annealing at 55 °C for 30 sec, extension at 72 °C for 30 sec, and final extension at 72 °C for 5 min. The 16S amplicons were purified using AMPure XP beads (Beckman Coulter, Brea, CA, USA), and paired-end libraries were attached using the Nextera XT Index Kit (Illumina, San Diego, CA, USA) and Illumina sequencing adapters (Illumina, USA). Indexed amplicons were cleansed with AMPure beads and normalised with PhiX Control v3 (Illumina, USA). Purified PCR amplicons were then sequenced using the Miseq platform (Illumina, USA).

### 2.6. Data Analysis

Raw reads were quality controlled and filtered (Q > 20 and length > 50 bp) using fastqc (v0.11.8) and trimGalore (v0.6.4_dev; https://github.com/FelixKrueger/TrimGalore, accessed on 13 July 2022), respectively. TrimGalore was also used to remove adapters. Krona charts for interactive data visualisation were generated using Kraken2 [26] and Krona [27]. All downstream analyses, including classification, abundance estimations, statistical analysis, and visualisation, were carried out in R (v3.6.1 Dada2 package (v1.12.1) [28] which was used to pre-process clean reads, including quality inspection, trimming, de-replication, merging paired-end reads, and removal of chimeric sequences. The obtained amplicon sequence variants (ASVs) were taxonomically classified, and ASV abundance estimates were determined using training sequence sets based on the SILVA reference database (v138; https://zenodo.org/record/1172783#.XvCmtkUzY2w, accessed on 14 July 2022). Ordinations for beta diversity, abundance bar plots, alpha diversity, and richness estimates were generated using the phyloseq (v1.28.0) [29], ggplot2 (v3.2.1), and AmpVis2 (v2.6.4) [30] packages. To compare alpha diversity between groups, Wilcoxon and/or Kruskal–Wallis rank-sum tests were used. The UpSet plots were generated using UpsetR (v1.4.0) [31]. DESeq2 (v1.24.0) was used to perform differential abundance analysis between sample groups [32].

## 3. Results

### 3.1. Alpha Diversity

A total of 397,766 quality raw reads were generated from all the samples collected (Table 2). The alpha diversity indices revealed comparably similar community diversity across the sites, except for WWTP4, which showed the highest diversity and species richness. The values for species richness (Chao1 and Abundance-based coverage estimators) ranged between 534–957 and for evenness (Shannon) ranged between 5.85–6.32. These results were confirmed by the rarefaction curve estimates, with WWTP4 presenting considerably higher values than the other sites, followed by relatively similar low values between WWTP1 and 2, and WWTP 3 and 5.

The rarefaction curves for the observed operational taxonomic units (OTUs) for all samples plateaued at around 5000 reads (Figure 1), indicating adequate sequencing depth; hence, a good representation of the bacterial community was achieved as most of the abundant species and some rare species are represented.

### 3.2. Taxonomic Comosition at Phylum, Family, and Genera Levels

A total of 24 phyla were identified, with fourteen (*Acidobacteriota, Actinobacteriota, Bacteroidota, Campilobacterota, Cyanobacteria, Desulfobacterota, Elusimicrobiota, Firmicutes, Fusobacteriota, Patescibacteria, Proteobacteria, Spirochaetota, Synergistota, Verrucomicrobiota*) appearing across the five WWTPs (Figure 2). Of these, the five most predominant phyla were *Bacteroidota* (31.7%), *Campilobacterota* (18.4%), *Proteobacteria* (20.2%), *Firmicutes* (8.6%), and *Desulfobacterota* (6.5%).

A total of 122 families were identified across all samples. Families of the top three dominant phyla contributing at least 1% to total abundance in the five WWTPs influent samples are shown in Figure 3. For the phylum *Bacteroidota*, the predominant families were *Bacteroidaceae*, *Paludibacteraceae*, *Prolixibacteraceae*, *Tannerellaceae*, and *Williamwhitmaniaceae*. The phylum *Proteobacteria* was dominated by *Pseudomonadaceae*, and *Rhodocyclaceae* and *Campilobacterota* were dominated by *Arcobacteraceae* and *Sulfurospirillaceae*.

At the genus level, 253 genera were identified from all samples accounting for the majority of the classified sequences, with approximately 25% of the total communities remaining unidentified. Genera that were detected at all sites with a relative abundance of at least 1% are presented in Figure 4. *Sulfurospirillum*, *Macellibacteroides*, and *Bacteroides* were the three most abundant genera identified from the influent samples, accounting for 12.8, 5.2, and 5.1% of total genera, respectively.

### 3.3. Shared and Distinct Bacterial Genera

Shared bacterial genera (those that appeared in two or more WWTPs) and distinct genera (those that occurred at one WWTP) are presented in Figure 5. Of the 253 genera identified, 40 were shared by all five sites, 24 shared by four sites, 32 by three sites, and 29 by two sites. WWTP4 had the highest bacterial richness while WWTP3 showed the least richness in the number of genera detected. We also observed genera that were unique to specific sites with 62 distinct genera observed at WWTP4, 29 at WWTP1, 15 at WWTP2, and 14 at WWTP5. The least diverse bacterial genera, with only eight distinct genera, were observed in WWTP3.

### 3.4. Genera That Contain Potential Pathogenic Species in Influent Samples

A total of 36 genera (approximately 20% of total abundance) of medical importance to human health were identified (Table 3) and belonged to the major phyla *Bacteroidota*, *Campilobacterota, Proteobacteria, Firmicutes, Synergistota, Fusobacteriota*, and *Actinobacteriota*. The dominant pathogenic genera were *Bacteroides* (5.1%), *Pseudomonas* (2.9%), *Aeromonas* (2.8%), *Arcobacter* (2.6%), *Leptotrichia* (1.4%), *Treponema* (0.9%), *Streptococcus* (0.6%), *Enterobacter* (0.5%), *Shewanella* (0.5%), and *Acinetobacter* (0.4%).

#### Pathogenic Genera and Their Potential Health Outcomes

Table 3 shows the relative abundance of pathogenic genera from the bacterial community as represented by influent samples of the five WWTPs. Potentially pathogenic bacterial genera were classified into three main groups according to the type of infection they cause in humans: (1) respiratory pathogens, (2) enteric pathogens, and (3) opportunistic pathogens commonly associated with nosocomial infections and multidrug resistance.

**Respiratory:***Mycobacterium* and *Coxiella* were the most abundant respiratory tract-associated pathogens contributing up to 0.1% and 0.05% of the total bacterial community of influents, respectively. *Mycobacterium* was detected only at WWTP1 and WWTP2, with a relative abundance of 0.2% at each plant, while *Coxiella* was only detected at WWTP4, accounting for approximately 0.2% of the total bacterial community at this site.

**Enteric:** Enteric pathogens detected were *Escherichia/Shigella*, *Laribacter*, *Arcobacter*, and *Aeromonas* with total relative abundances of 0.1%, 0.2%, 2.6%, and 2.8%, respectively. *Aeromonas* was the most prevalent enteric pathogen, with a relative abundance ranging between 0.7% and 5.3%, followed by *Arcobacter* with the highest abundance (9.7%) at WWTP5 and the lowest at WWTP3 (0.9%). It is also important to mention that *Aeromonas* and *Arcobacter* were detected at all sites. *Escherichia/Shigella* were rare genera detected only at WWTP1 and WWTP2, with abundances of 0.3% and 0.1% at the two sites, respectively.

**Opportunistic:** Thirty opportunistic bacterial genera were identified in the present study, with an overall abundance of 14.4% in the total bacterial community (Table 3). The top three opportunistic pathogens were *Bacteroides, Pseudomonas*, and *Leptotrichia* contributing (5.1%, 2.9%, and 1.4%, respectively, of the total bacterial community. The other opportunistic genera were ˂1% in relative abundance. A total of 13 opportunistic genera (*Acinetobacter, Aeromonas, Arcobacter, Bacteroides, Comamonas, Dysgonomonas, Enterobacter, Leptotrichia, Pseudomonas, Pseudoxanthomonas, Shewanella, Streptococcus*, and *Treponema*) were detected in at least four of the five WWTPs.

### 3.5. Risk Characterisation of Potentially Pathogenic Bacteria

The observed pathogenic genera were classified into different risk groups according to the revised South African Regulation for Hazardous Biological Agents, 2022 [33] (Table 3). Thirteen of the 36 genera (36%) belong to HBA Risk Group 2 (may cause disease, are unlikely to spread to the community, and effective treatment is available), and three genera belong to HBA Group 2 or 3 (HBA Risk Group 3, may cause severe disease, present a risk of spreading to the community but effective treatment is available) depending on the species type (Table 3). *Coxiella* belongs to HBA Risk Group 3 and has only one member, *C. burnetii*. In summary, close to 50% (17/36) of pathogenic genera identified were classified as hazardous biological agents, indicating that these organisms can cause human diseases and may pose a health risk to WWTP workers.

## 4. Discussion

### 4.1. Bacterial Community Composition in Influent Samples

Municipal WWTPs receive wastewater from different sources but mostly households, surface runoff, and industrial activities, contributing to the complexity of bacterial communities in wastewater [34]. The phyla *Bacteroidota*, *Campilobacterota*, *Proteobacteria*, *Firmicutes*, and *Desulfobacterota* were predominant in the present study. Except for *Campilobacterota*, the top dominant phyla in the current study have previously been reported in high abundance in influent wastewater [35,36]. Wu and colleagues (2019) [37] recently provided a comprehensive wastewater analysis on a global scale, representing 23 countries from six continents including Africa, Asia, Australia, Europe, North America, and South America. Their findings revealed that despite the considerable diversity in bacterial communities between samples, a core global community (28 OTUs) exists. A majority of these members belonged to the phyla *Proteobacteria* and *Bacteroidota*, implying some degree of bacterial community conservation in municipal wastewater at higher taxa rank [37]. In addition, the phyla *Bacteroidota*, *Proteobacteria*, *Firmicutes*, and *Actinobacteria* have consistently been reported as the predominant bacterial community members in the human microbiome, suggesting that a large proportion of bacterial members in the samples analysed originated from human faecal material [38]. When comparing bacterial communities in other wastewater types, in particular, activated sludge, previous studies highlighted a distinct microbial ecosystem with high bacterial diversity and a high concentration of biomass in activated sludge samples [39,40]. Begmatov and co-workers [40] recently reported a dominance of *Proteobacteria*, *Chlorofexi*, *Myxococcota*, *Firmicutes*, *Patescibacteria*, and *Nitrospirota* in activated sludge [40]. A comparison between the activated sludge community identified in Moscow with that identified globally by the Global Water Microbiome Consortium show clustering, emphasising that influent characteristics, which are largely influenced by cultural, social, and environmental factors in each region, are more important than WWTP operating conditions [40].

It is noteworthy that at the genus level in the current study, bacterial composition exhibited some degree of diversity across the five WWTPs. The highest bacterial richness was recorded at WWTP4, whereas WWTP3 showed the least richness in the number of genera detected. The lowest bacterial richness was found at WWTP3, which could be explained by the fact that this plant also treated industrial wastewater, whereas the other four WWTPs received municipal wastewater primarily from households. Chemical substances in industrial wastewater have been shown to negatively impact microbial community structures, resulting in reduced bacterial richness and diversity compared to municipal wastewater [41,42]. Interestingly, WWTP4 contained many distinct genera compared to the other plants, which may be attributed to it being the largest treatment plant in the area, serving approximately seven different communities. Overall, these findings suggest that while some WWTPs harboured exclusive genera, the predominant genera did not vary considerably regardless of plant location, indicating a high degree of bacterial community stability in the influent source.

### 4.2. Potentially Pathogenic Genera

This study grouped bacterial genera with known disease-causing species into three major infection categories: respiratory, enteric, and opportunistic pathogens.

*Coxiella* and *Mycobacterium* were the only medically important genera identified in the present study with the potential of causing respiratory tract infections. Although the total relative abundance of these genera was much lower than that of the majority of the identified genera, *C. burnetii* and *M. tuberculosis* have extremely low infectious doses, requiring less than 10 living organisms to cause an infection [43,44]. Respiratory pathogens cause infections of the upper and lower respiratory tract. By far, the most serious respiratory infections involve the lower respiratory tract such as bronchitis and pneumonia, and are the leading cause of high mortality rates worldwide [45]. The abundance of *Mycobacterium* in influent samples was comparable to previous studies [10,12,46]. For instance, studies conducted in Germany and Australia found that the overall abundance *Mycobacterium* in influent and effluent was less than 0.02% [12,46]. In comparison to other types of wastewater, studies on *Mycobacterium* in influent are not common. However, *Mycobacterium* has primarily been studied in effluents to assess the efficacy of wastewater treatment processes in removing biological agents and the safety of treated effluents [21,36,47]. Humans become infected with *Coxiella* by inhaling aerosols from contaminated animal waste, soil, or food products, and veterinarians, slaughterhouse, and farm workers are generally considered to be at increased risk of occupational exposure to *Coxiella* [43]. Given that WWTPs receive wastewater from various institutions and farms, it raises the question of whether WWTP workers are at risk of occupational exposure to this pathogen.

Five enteric genera, namely, *Escherichia/Shigella*, *Laribacter*, *Arcobacter*, and *Aeromonas* were identified in the study, accounting for 5.7% relative abundance of the total bacterial community. Enteric pathogens normally reside in the intestines of humans and can utilise their pathogenic mechanisms to cause gastrointestinal tract infections [48]. Enteric organisms are typically transmitted via the faecal–oral route, and illness symptoms are caused by consuming contaminated food or water [48]. Using metagenomics analysis, previous studies have reported the prevalence of *Aeromonas* to be noticeably high in wastewater with counts similar to those of faecal coliforms [47,49,50]. Ye and co-workers [47] analysed potentially pathogenic bacteria in sixteen samples comprising of influent, activated sludge, and effluent from 14 municipal WWTPs across, China, Canada, United States, and Singapore, and reported that *Aeromonas* were among the most dominant genera occurring at least in ten samples across four countries [47]. Other metagenomics studies [17,51,52,53] have also reported a high abundance of *Aeromonas* in various wastewater types. *Aeromonas* genus has been ranked third as the leading cause of diarrhoea after *Camplylobacter* and *Salmonella* [54], and incidences of gastroenteritis linked to *Aeromonas* have been reported across the globe, with cases more common in developing countries [54]. Two *Arcobacter* species, *A. butzleri* and *A. cryaerophilus*, are considered emerging pathogens threatening human health [55]. Human-associated *Arcobacter* species have been consistently isolated from human sewage systems [35,55]. In the present study, the genus *Arcobacter* was observed in high abundance exclusively at WWTP5 (9.7%), compared to the other WWTPs that had an abundance ranging from 0.9% to 2.5%. This inconsistency could not be explained further. In the present study, the genus *Laribacter* was 0.2% relative abundance, which is comparable with that reported in municipal influent (≤0.1% relative abundance) from a study conducted at three WWTPs in Western Australia [56]. The genus *Laribacter* has one species, *L. hongkongensis*, which is associated with traveller gastroenteritis and diarrhoea [57]. It should be noted that, even though *Laribacter* has not been previously reported among the most prevalent genus in wastewater, if ingested, this bacterium can cause gastroenteritis in humans [56].

The present study identified 32 genera that could cause opportunistic infections in humans. Although opportunistic pathogens pose little risk to healthy WWTP workers, they can cause serious illnesses in individuals with weakened immune systems and the elderly. The main concern with opportunistic organisms is that they are typically resistant to commonly used antimicrobial treatments, posing a serious problem for public health [58]. Five pathogenic genera (*Enterococcus*, *Klebsiella*, *Acinetobacter*, *Pseudomonas*, and *Enterobacter*) that could contain species belonging to the group of ESKAPE pathogens were identified in the present study with *Pseudomonas* topping the list. Members of the ESKAPE pathogens are well known for their ability to develop multidrug resistance and account for most nosocomial infections [59]. Moreover, *Acinetobacter baumannii*, *Pseudomonas aeruginosa*, and *Streptococcus pneumonia* are on the WHO priority list of antibiotic-resistant pathogens as they can cause fatal infections [60]. Resistance to commonly used antibiotics is a serious global problem. Current efforts to monitor the development of antimicrobial resistance have mainly focused on clinical settings; however, interest has grown in recognising the importance of antibiotic resistance in the environment and water supply [57]. In fact, WWTPs have been implicated as key reservoirs, incubators, and source for disseminating antimicrobial resistance and virulence genes [61].

### 4.3. Risk Characterisation of Identified Potential Pathogens

In this study, 17 genera were classified as potentially hazardous biological agents to human health. Overall, the risk characterisation exercise revealed that workers at WWTP may be exposed to genera that cause airway obstruction, gastrointestinal problems, and opportunistic infections in the workplace. Except for *Mycobacterium* and a few Gram-positives such as *Streptococcus*, *Enterococcus*, *Leuconostoc*, *Erysipelothrix*, *Atopobium*, *Brachybacterium*, *Finegoldia*, *Gordonia*, and *Actinomyces*, a majority (28/36 genera) of the classified pathogenic genera identified at WWTPs were Gram-negative. Gram-negative bacteria express endotoxins as their main component of the outer membrane [62], and the presence of numerous and diverse Gram-negative bacteria at WWTPs may be a major contributor to workers’ exposure to elevated levels of endotoxins. Long-term exposure to inhalable endotoxins has been linked to inflammatory responses in the lungs, leading to symptoms such as chronic bronchitis, organic toxic dust syndrome, or asthma [63]. Therefore, the findings of this study suggest that workers at the selected five WWTPs may be exposed to pathogens, including endotoxins from Gram-negative bacteria, which could compromise their respiratory health.

Waterborne diseases are expected to rise with climate change and a growing global population [64]. Consequently, WWTP workers remain at risk from waterborne diseases and outbreaks irrespective of the country’s economic status (i.e., developed or developing) or whether in the tropics or temperate, highlighting the importance of regular monitoring of wastewater microbiomes using advanced but cost-effective detection techniques, precise disinfectant procedures, and proper management of operations to minimise worker exposure. The morbidity and mortality of waterborne diseases are enormous [64], and they can only be controlled by providing workers with a microbiologically safe environment. WHO has established a list of priority waterborne pathogens with moderate to high health significance [65]. In the present study, *Mycobacterium* and *Escherichia/Shigella* genera were detected in wastewater and these organisms are listed on the WHO priority pathogens. Additionally, emerging waterborne pathogens of importance *Aeromonas* and *Leptotrichia* were detected in the study. Therefore, the findings of this study warrant further investigation into incidences of infections in WWTP workers associated with exposure to these pathogens. Workers at WWTP play an important role in urban communities, making sure that the treatment plants function optimally. However, the working conditions for sanitation workers, including WWTP workers, expose them to significant health risks such as waterborne diseases, injuries, and even death [22].

### 4.4. Limitations of the Study

Data presented in this study are from a single type of wastewater (influent), and samples were collected once over one month from five municipal treatment plants in South Africa. Hence, the small sample size is a limitation. Furthermore, sample collection during and outside the respiratory infection season was impractical as some COVID-19 symptoms were similar to respiratory influenza and occurred throughout the year; hence, defining the respiratory infection season was challenging and thus omitted. Therefore, more research is needed to monitor trends and changes in the diversity of bacterial pathogens in various types of wastewater on a large scale over a long period to assess temporal and spatial variation. Although the current study provides local baseline data on potential pathogens circulating in WWTP environments, further research to provide impact of exposure on workers’ health is needed. Therefore, a follow-up study assessing the associations between work activities and the incidence of gastrointestinal and respiratory infections among WWTPs in South Africa is underway. This information is currently non-existent in South Africa, making it difficult to reform occupational health and safety policies and practices.

## 5. Conclusions

The study found evidence of several potentially pathogenic bacteria in untreated municipal wastewater, which may pose a health risk to WWTP workers. The major phyla identified in the study were *Bacteroidota*, *Proteobacteria*, *Campilobacterota*, *Firmicutes*, and *Desulfobacterota* and the dominant pathogenic genera were *Bacteroides*, *Pseudomonas*, and *Aeromonas*. Risk characterisation of the identified pathogenic genera revealed that the assigned genera are capable of causing gastrointestinal illnesses, airway obstruction, and some are intrinsically resistant to commonly used antibiotics. However, further studies are imperative to establish an association between the identified pathogenic bacterial pathogens and the commonly reported symptoms among WWTP workers. Despite the preliminary nature of the results, intervention strategies should focus on raising awareness of bacterial contaminants present at WWTP and improving personal protective equipment (PPE) compliance in workers to mitigate health risks.

WWTP workers are vulnerable to increased occupational and environmental health hazards, barriers to healthcare, access to personal protective equipment, legal protection, and other safeguards. Molecular profiling of bacterial communities generates scientific-based evidence on the abundance of potential pathogenic bacteria in untreated wastewater that WWTP workers may be exposed to. With a lack of such information for workers in LMICs, comparison with high income countries is impracticable. Therefore, the findings of this study contribute to the body of knowledge, the relevance of region-specific data for a low-income countries given that bacterial communities in wastewater differs between geographical locations and are influenced by factors such as cultural, social, and environmental conditions. This may lead to the identification of region-specific diseases thus tailoring interventions that are specific and targeted to local settings.

## Figures and Tables

**Figure 1 ijerph-20-04338-f001:**
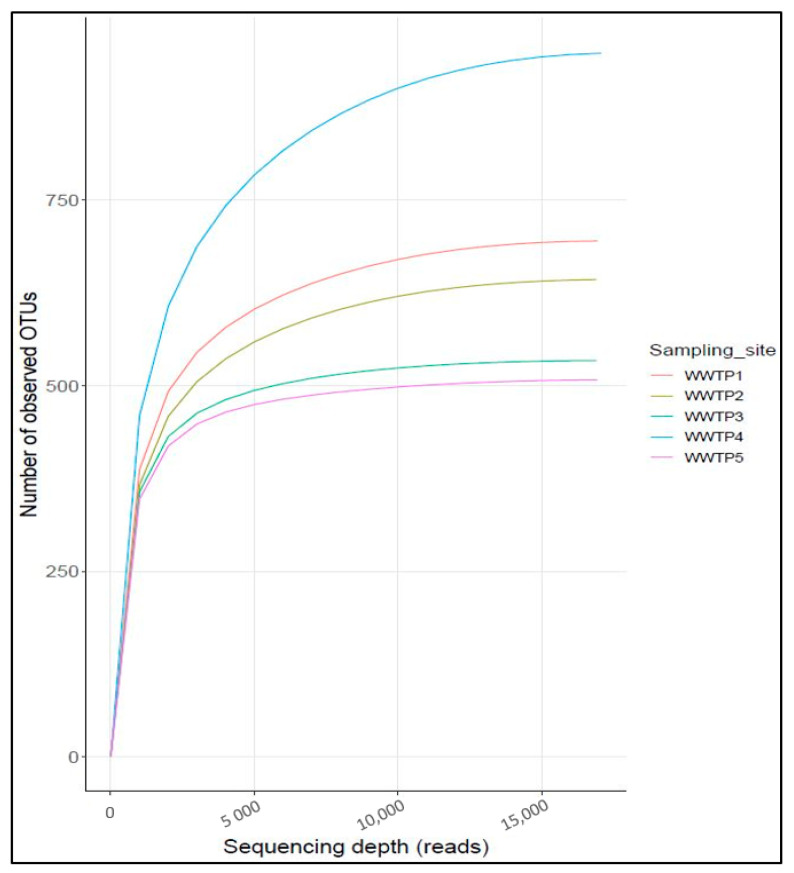
Rarefaction curves showing bacterial richness of influent wastewater from five WWTPs.

**Figure 2 ijerph-20-04338-f002:**
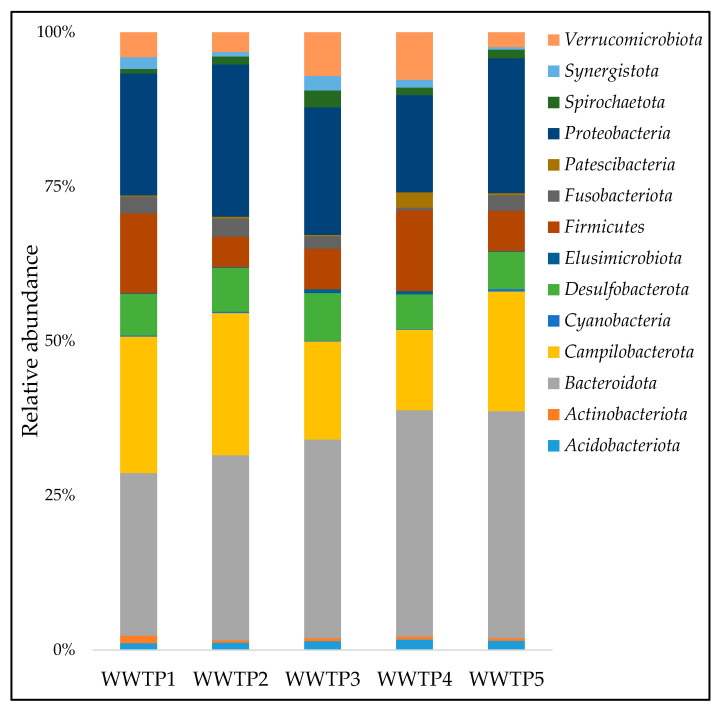
Relative abundance of the predominant phyla in influent wastewater.

**Figure 3 ijerph-20-04338-f003:**
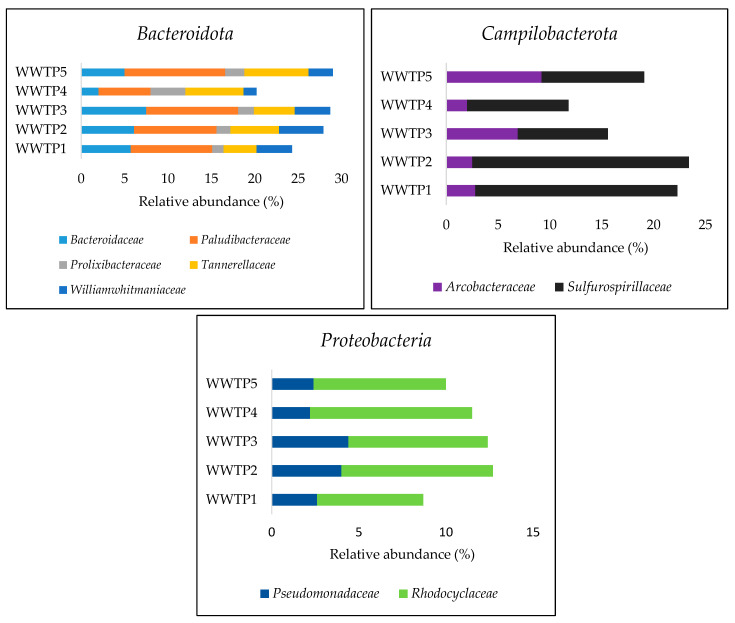
Relative abundance of the families of the top three dominant phyla *Bacteroidota*, *Campilobacterota*, and *Proteobacteria* from the WWTP influent samples.

**Figure 4 ijerph-20-04338-f004:**
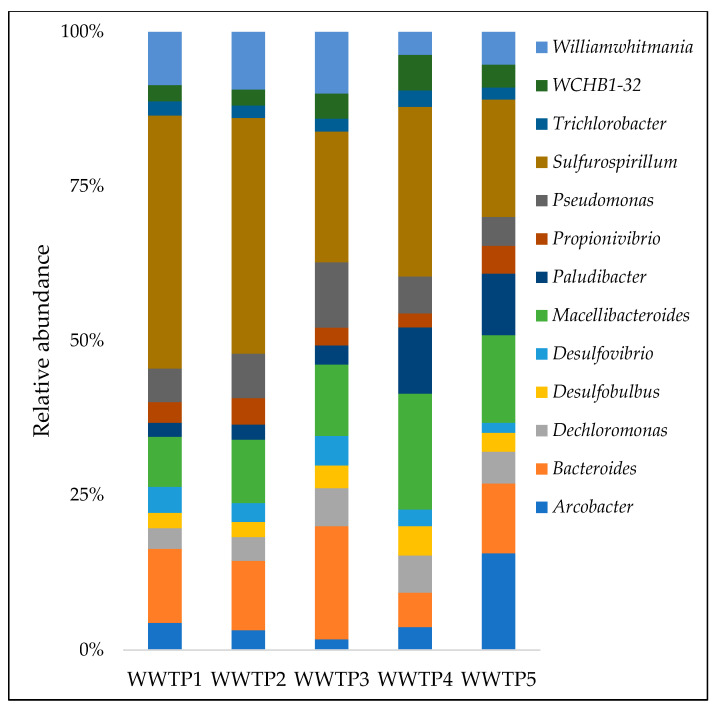
Relative abundance of predominant genera across the five WWTPs.

**Figure 5 ijerph-20-04338-f005:**
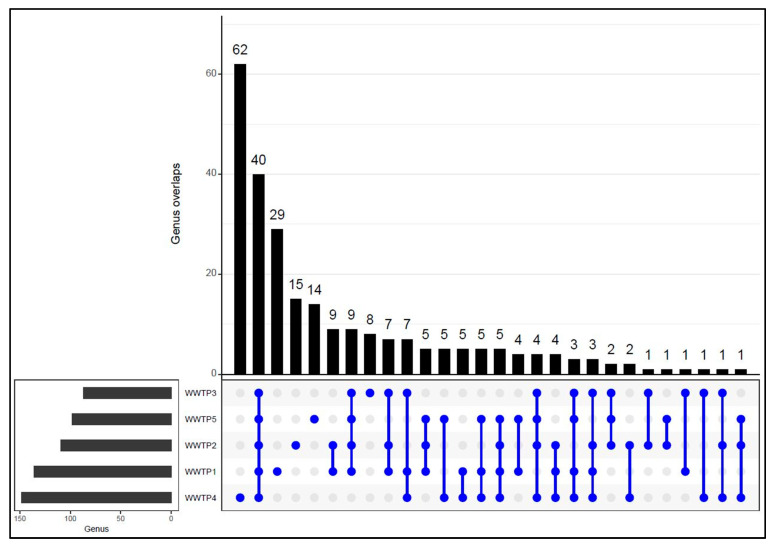
UpSet plot showing shared and distinct bacterial genera in influent wastewater from five WWTPs. Numbers above the bars indicate the number of overlapping genera between different WWTPs appearing in the left panel display.

**Table 1 ijerph-20-04338-t001:** Characteristics of WWTP sampling sites.

Site	WWTP1	WWTP2	WWTP3	WWTP4	WWTP5
Source of wastewater (%)	Mixed (domestic (90) and industrial (10)	Domestic (100)	Mixed (domestic (80) and industrial (20)	Domestic (100)	Domestic (100)
Population size served	366,709	600,000	236,580	1,041,200	472,000
Treatment capacity * (ML/day)	35	60	93	180	85
Treatment train	Raw influent, bar screens, grit removal chamber, primary clarifiers, surface aeration tank, secondary sedimentation	Raw influent, bar screens, grit removal chamber, primary clarifiers, surface aeration tank and trickling bio-filters, secondary sedimentation	Raw influent, bar screens, grit removal chamber, primary clarifiers, surface aeration tank, secondary sedimentation	Raw influent, bar screens, grit removal chamber, primary clarifiers, diffused aeration tank, secondary sedimentation	Raw influent, bar screens, grit removal chamber, primary clarifiers, diffused aeration tank, secondary sedimentation
Biological treatment	Activated sludge	Activated sludge and bio-filters	Activated sludge	Activated sludge	Activated sludge
Tertiary treatment	Chlorine disinfection	Chlorine disinfection	Chlorine disinfection	Chlorine disinfection	Chlorine disinfection
Intended reuse of treated effluent	Irrigation and housekeeping purposes	Irrigation and cooling water to a nearby power station	Housekeeping purposes	Discharged into a river	Agricultural purposes
Workforce	22	38	37	66	27

* ML/day—mega litres per day.

**Table 2 ijerph-20-04338-t002:** Community diversity of influent wastewater samples from five WWTPs.

Sample	Total Reads	ASV	Chao1	ACE	Shannon	Simpson
WWTP1	85,200	19,449	695	697	6.00	0.997
WWTP2	75,629	16,334	643	646	5.91	0.996
WWTP3	59,881	13,594	534	535	5.85	0.996
WWTP4	84,585	20,300	949	957	6.32	0.998
WWTP5	92,471	16,923	508	510	5.85	0.996

ASV: Amplicon sequence variant, ACE: Abundance-based coverage estimators.

**Table 3 ijerph-20-04338-t003:** Overall relative abundance of pathogenic genera and their risk evaluation by type of infection.

Genera by Type of Infection	Relative Abundance (%)	HBA Risk Group
**Respiratory:**		
*Coxiella*	0.05	3
*Mycobacterium*	0.1	2/3
**Enteric:**		
*Aeromonas*	2.8	2
*Arcobacter*	2.6	unclassified
*Escherichia/* *Shigella*	0.1	2/3
*Laribacter*	0.2	2
**Opportunistic:**		
*Acinetobacter*	0.4	2
*Actinomyces*	0.1	2
*Atopobium*	0.05	unclassified
*Bacteroides*	5.1	2
*Blastomonas*	0.05	unclassified
*Brachybacterium*	0.05	unclassified
*Chryseobacterium*	0.05	unclassified
*Citrobacter*	0.05	unclassified
*Comamonas*	0.3	unclassified
*Dysgonomonas*	0.3	unclassified
*Empedobacter*	0.05	unclassified
*Enterobacter*	0.5	2
*Enterococcus*	0.05	2
*Erysipelothrix*	0.1	2
*Finegoldia*	0.05	unclassified
*Gordonia*	0.05	unclassified
*Klebsiella*	0.05	2
*Leptotrichia*	1.4	unclassified
*Leuconostoc*	0.05	unclassified
*Ochrobactrum*	0.05	unclassified
*Prevotella*	0.2	2
*Pseudomonas*	2.9	2/3
*Pseudoxanthomonas*	0.1	unclassified
*Roseomonas*	0.05	unclassified
*Shewanella*	0.5	unclassified
*Sphingobacterium*	0.05	unclassified
*Streptobacillus*	0.05	2
*Streptococcus*	0.6	2
*Synergistes*	0.1	unclassified
*Treponema*	0.9	2

## Data Availability

Data are available upon request and within the prescripts of the Protection of Personal Information Act (POPIAct).

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
