# Peer review of "Potential Exposure to Respiratory and Enteric Bacterial Pathogens among Wastewater Treatment Plant Workers, South Africa"

_ijerph, 2023, doi:10.3390/ijerph20054338_

Round 1

Reviewer 1 Report

Line 31 -Introduction: Describe more about waterborne diseases and the risk to workers

Line 85 - Study design: It would be important to describe the location of collection points. The text only mentions that it is in a big city but does not say which one. It is necessary to include the region, describe the surrounding area and in the discussion explore the data found at different collection points with the population and surrounding region.

Line 91 Site description: Table 1. Characteristics of WWTP sampling sites need to improve with the region name of sites and the name of the city.

Line 211 Figure 5: the quality of the figure is not good. The design and content is good but the quality is untwisting the words and numbers.

Line 261 Discussion: Even understanding that the present study has limitations and that it does not propose to compare clinical data of workers with data found on the diversity of bacteria, I suggest adding one more item to the discussion in relation to waterborne diseases to enrich the discussion.

Line 386 Limitations of the study: Yes, i agree if this limitations. The work needs to be expanded not only by increasing the number of samples and collection points but also collecting clinical data from workers at each site and also doing a risk analysis with the data that already exists.

Reviewer 2 Report

General

The paper, titled "Potential Exposure to Respiratory and Enteric Bacterial Pathogens among Wastewater Treatment Plant Workers, South Africa", presents the bacterial diversity and pathogens that could pose occupational health risks from exposure to untreated municipal wastewater. Results were obtained by targeted 16 rRNA gene amplicon sequencing. The influent samples were obtained by single sampling from five participating wastewater treatment plants in a large South African metropolitan area.

Introduction

The introduction is mainly well written and provides background information on the topic based on relevant publications. The authors' rationale for conducting this study is that there is a general lack of research in low- to middle-income countries (LMICs) on characterizing bacterial communities in untreated wastewater using high-throughput sequencing technologies and that previous studies in South Africa have focused primarily on pathogen removal and wastewater quality. The authors do not assume possible reasons why the microbiome in the effluents of the studied WWTPs might differ from the microbiome described in previous publications.

Materials and Methods

The materials and methods are generally well written. The authors used current methods and appropriate materials in conducting the study. The statistical data processing is well done.

I miss some important details on study design. More information about the individual WWTPs might be useful to support/explain the differences in results. For example, is it possible to get information on which WWTPs receive effluent from the largest hospitals in the region, livestock farms, etc.?

In addition, the authors themselves have pointed out the limitations of the study, "Samples were collected for one month, and the small sample size is a limitation". The study would have been much better if the samples had been collected at least twice, if possible during and outside the respiratory infection season. This is a major limitation of the study.

Although the objective of the study is not to determine the actual impact of work in wastewater treatment plants on the incidence of infections, it would be useful to determine the proportion of worker absenteeism attributable to illnesses related to possible workplace infections. Otherwise, this study is not very scientifically sound, adds nothing new, and is merely a screening of the microbiome of an area's wastewater.

Results

Given the study design, the results are clearly presented. Reference to supplementary table S1 is missing (only in Materials and Methods - line 102). The resolution of Figure 1 should be improved.

Discussion

If it is possible to improve the study with the information suggested for the study design, the results and discussion should also be expanded according to the new additional data.

Lines 327-334 „In the present study, the genus Arcobacter was observed in high abundance exclusively at WWTP5 (9.7%), compared to the other WWTPs that had an abundance ranging from 0.9% to 2.5%. This inconsistency could be attributed to the environmental dynamics of wastewater systems. The complex sewage ecosystem has been suggested to play a role in Arcobacter growth favoring different Arcobacter species at different times [48]. Temperature, in particular, appears to play an important role in the type of Arcobacter species and quantity identified as some species may only be isolated from samples with low temperatures (Ë‚20°C), resulting in seasonal changes in Arcobacter composition in wastewater “.

Are temperature differences of only 1-2 °C (Table S1) really significant enough to be the reason for significant differences in Arcobacter abundance among WWTP-s? If it is not due to temperature alone, then perhaps it is due to the environmental dynamics of the wastewater system, as suggested by the authors. If so, then a description of the wastewater system in Materials and Methods should be added, as I suggested above.

Although the conduct of the study is explained by the lack of such studies in areas in low to middle income countries, a clear comparison with other studies, particularly with studies in higher income countries is lacking. It lacks a clear conclusion as to why this study is relevant from a scientific perspective.

Round 2

Reviewer 2 Report

I carefully read the new version of the manuscript and the author's comments. The authors took into account all my comments and improved the manuscript considerably. Although they could not improve the manuscript in some segments, they communicated it clearly. There are still some minor details to solve, e.g. Fig. 1 is missing (probably they forgot to add it) and in Fig. 5 the resolution needs to be improved.
